# Trabecular Bone Score Preceding and during a 2-Year Follow-Up after Sleeve Gastrectomy: Pitfalls and New Insights

**DOI:** 10.3390/nu15153481

**Published:** 2023-08-07

**Authors:** Joshua Stokar, Tair Ben-Porat, Donia Kaluti, Mahmud Abu-Gazala, Ram Weiss, Yoav Mintz, Ram Elazari, Auryan Szalat

**Affiliations:** 1Osteoporosis Center, Endocrinology and Metabolism Service, Internal Medicine Ward, Hadassah Medical Center, Faculty of Medicine, Hebrew University of Jerusalem, Jerusalem 9124001, Israel; 2Department of Human Metabolism and Nutrition, Braun School of Public Health, Hebrew University, Jerusalem 9124001, Israel; tairbp20@gmail.com (T.B.-P.); doniak@hadassah.org.il (D.K.); 3Department of Nutrition, Hadassah-Hebrew University Medical Center, Jerusalem 9124001, Israel; 4Department of Health, Kinesiology and Applied Physiology, Concordia University, Montreal, QC H4B 1R6, Canada; 5Department of Surgery, Hadassah Medical Center, Faculty of Medicine, Hebrew University of Jerusalem, Jerusalem 9124001, Israel; mabugazala@hadassah.org.il (M.A.-G.); ymintz@hadassah.org.il (Y.M.); ramelazary@hadassah.org.il (R.E.); 6Department of Pediatrics, Rambam Medical Center, Technion School of Medicine, Haifa 3200003, Israel; ramw@rambam.health.gov.il

**Keywords:** trabecular bone score, sleeve gastrectomy, dual-energy X-ray absorptiometry, body mass index, waist circumference, tissue thickness

## Abstract

Bariatric surgery (BS) can have negative effects on bone health. Bone microarchitecture quality evaluation using the trabecular bone score (TBS) has not been described in patients after sleeve gastrectomy (SG). To test the hypothesis that the TBS is clinically useful for this population, we evaluated changes in bone mineral density (BMD) and the TBS in a longitudinal cohort study following SG. The measurements before surgery and after 12 and 24 postoperative months were as follows: weight, height, BMI, waist circumference (WC), BMD and TBS. The results at baseline showed the following: a mean BMI of 43 ± 0.56, TBS of 1.25 ± 0.02, lumbar spine BMD T-score of −0.4 ± 0.93, TBS T-score of −2.30 ± 0.21, significantly lower than BMD-T-score, and associated with a BMD-T-TBS-T gap (T-gap) of −2.05 ± 1.26 (−0.24 ± 0.13). One year after surgery, the TBS had significantly improved (+12.12% ± 1.5), leading to a T-gap of −0.296 ± 0.14, which remained stable at 2 years post-surgery. A correlation analysis revealed a significant negative correlation between the T-gap and WC (r = −0.43 *p* = 0.004). Our interpretation is that abdominal fat may interfere with image acquisition via increased tissue thickness, leading to a false low TBS at baseline. In conclusion, TBS should be interpreted with caution in patients with obesity and elevated WC. Additionally, we show that after SG, the LS microarchitecture measured using the TBS is partially degraded in up to 25% of patients. Further studies are warranted to assess hip bone microarchitecture changes after bariatric surgery.

## 1. Introduction

Bariatric surgery (BS) is the most efficient treatment for weight loss and the improvement of obesity-related comorbidities such as diabetes mellitus, dyslipidemia and hypertension, as well as reducing mortality in comparison to matched non-surgical patients [1].

However, weight loss secondary to Roux-en-Y gastric bypass (RYGB) or sleeve gastrectomy (SG) has been shown to have a negative effect on bone health, with an increased incidence of osteopenia and osteoporosis [2]. Various mechanisms for this effect have been suggested: a low dietary intake, muscle and skeletal unloading, the malabsorption of nutrients and changes in gut hormone secretion [3,4]. Osteoporosis is associated with an increased risk of fractures due to the loss of bone strength secondary to low bone mineral density (BMD) and/or microarchitectural abnormalities [5,6]. Both these components of bone strength are impacted by bariatric surgery [7]. BMD, as evaluated via dual-energy X-ray absorptiometry (DXA), decreases significantly in the first few years after BS. Bone microarchitecture, as assessed using high-resolution peripheral quantitative computerized tomography of the distal radius and tibia, which evaluates volumetric trabecular and cortical bone, is significantly impaired early on following BS. The trabecular bone score (TBS) is another method used to evaluate bone microarchitecture, based on a gray-scale texture measurement obtained from the spine DXA image [8]. A TBS above 1.3 indicates normal bone microarchitecture, a score between 1.2 and 1.3 indicates partially degraded bone microarchitecture, and a score below 1.2 indicates degraded microarchitecture. TBS values are also converted into TBS-adjusted T-scores to complement BMD-based T-scores. However, BMD-based T-scores remain the gold standard for diagnoses based on the World Health Organization (WHO) classification: normal if the BMD T-score is >−1.0; osteopenia for a <−2.5 < BMD T-score < −1.0; and osteoporosis if the BMD T-score is <−2.5. There are clear advantages of using the TBS in addition to DXA to evaluate patients’ fracture risk, as it is not affected by degenerative spine changes, and its use has been validated in many clinical circumstances [9]. The use of the TBS is limited to patients with a body mass index (BMI) between 15 and 37 kg/m^2^. Thus, in patients with a BMI above 37 kg/m^2^, the role of the TBS in their bone health assessment has not been validated. This limitation is relevant to most patients undergoing BS. However, the changes in TBS monitoring after bariatric surgery may be of interest. The TBS was shown to be maintained at 1 and 3 years in patients after RYGB [10]. To the best of our knowledge, no such study has been published based on patients after SG. We recently showed that among 33 patients who were previously on DXA follow-up, at 1 and 2 years after SG, bone mineral density (BMD) decreased significantly in the total hip (with an increase between 1 and 2 years), whereas it remained stable in the spine [11,12]. The main goal of our current investigation was to evaluate the hypothesis that the TBS has valuable clinical relevance in patients undergoing SG despite their elevated BMI before surgery and that TBS may be impacted by SG during follow-up. Herein, we offer an analysis of the microarchitectural changes in these patients using TBS monitoring at baseline before surgery 1 and 2 years post-surgery.

## 2. Methods

### 2.1. Study Overview

As previously described, 62 patients were enrolled in a prospective, randomized clinical trial (Identifier: NCT02483026) to compare the effect of pre-operative vitamin administration vs. standard pre-surgical care on BMD among female candidates for SG, with a follow-up at one [11] and two years after surgery [12]. The present longitudinal cohort study reports additional data concerning the bone health evaluation of these patients obtained at the time when the DXA scans were performed, namely, their trabecular bone score values. In total, 54 patients completed follow-up at 12 months. No differences were observed between the groups at 12 months of DXA follow-up [11], and thus, a combined group of 33 patients had an additional DXA scan performed at 24 months [12]. At baseline, the inclusion criteria were women aged 18–65 years with a BMI ≥ 40 kg/m^2^ or BMI ≥ 35 kg/m^2^ with comorbidities. Patients were excluded in cases of untreated mental illness or an unstable mental state, pregnancy, lactation, chronic conditions or medications affecting bone metabolism and previous BS. The study was approved by the institutional review board, and all the participants signed an informed consent form. 

### 2.2. Measures and Outcome Variables 

Between January 2018 and April 2021, after 12 and 24 postoperative months, participants were evaluated for their weight and height (measured on a digital medical scale and a stadiometer, respectively) at baseline, and their waist circumference was measured twice at the level of the umbilicus according to a uniform protocol. BMI was calculated based on weight (in kilograms) divided by height squared (in square meters). Patients received standard-international-guidelines-based post-bariatric medical care, nutritional counseling and routine supplementation [13,14] and were advised to take 3000 IU of vitamin D3 daily, 1200 mg of calcium from food and supplements, a standard multivitamin daily and a minimum daily protein intake of 60 gr/d from food post-surgery. 

DXA scans were performed using the same machine (Hologic Discovery, Hologic, Inc. Bedford, MA, USA) in an array mode at the lumbar spine (LS) (L1–L4), total hip and femoral neck, by the same experienced technician. Precision errors for areal BMD assessments were determined according to recommendations from the International Society for Clinical Densitometry [15]. The least significant change s for the lumbar spine, femoral neck, and total hip BMD were 0.36, 0.28 and 0.19 g/cm^2^, respectively. TBS was obtained from DXA scans using TBS Insight software version 3.03, following validated calibration with the TBS phantom as recommended by the manufacturer. TBS results were associated with a TBS-derived T-score allowing us to compare the latter with the DXA-related LS T-score and to calculate a BMD T-score–TBS T-score gap (T-gap). 

### 2.3. Statistical Analysis 

The statistics, using continuous variables, are presented as means ± SD. A paired comparison of continuous variables between 2 timepoints was performed using the paired *t*-test or mixed effects analysis for more than two groups with the Holm–Šídák correction for multiple comparisons. The Pearson correlation coefficient was calculated to assess the strength of the linear association between two continues variables. All tests applied were two-tailed, and a *p*-value of 5% or less was considered statistically significant. All the statistical analyses were performed using Graphpad Prism for Windows version 9.3.1.

## 3. Results

A total of 52 female patients with BMD and TBS measurements were available for analysis. Of those, 42 patients had TBS measurements prior to surgery, 48 at 1 year post-surgery and 31 at 2 years post-surgery. The clinical and anthropometric characteristics of the cohort are presented in Table 1.

The LS BMD measurements based on the WHO classification, as well as the TBS results and changes from baseline to 1 and 2 years after SG, are presented in Table 2. 

At baseline, the results suggest that bone microarchitecture evaluated using the TBS is more negatively impacted by obesity than BMD, as 26/42 (61.9%) of the patients had an abnormally low TBS (degraded or partially degraded), whereas only 9/52 (17.3%) of the patients had an abnormal BMD (osteopenia or osteoporosis). At 1 year, the LS BMD remained relatively stable and was abnormal in 10 patients (19.2%), but the TBS improved, as only 6 patients had an abnormal TBS (12.5%). Two years after SG, the LS BMD was abnormal in four patients (12.9%), and TBS was partially degraded in eight patients (25.8%).

Changes in the numbers of patients according to BMI categories (above 37, between 30 and 37, and below 30 kg/m^2^) at baseline one and two years after surgery are reported in Table 3. Obviously, before SG, most patients had a BMI > 37 kg/m^2^, which was significantly decreased at 1 and 2 years after SG; they represented only 20% of patients at the end of the follow-up. Between the first and second years, however, this proportion mildly increased.

The changes in the LS BMD, LS T-Score, TBS and TBS-adjusted T-score are shown in Table 4.

The mean pre-surgery difference between the BMD T-score and TBS T-score was −2.05 ± 1.26 (N = 42). One year after surgery, the LS BMD and T-scores remained relatively unchanged. In contrast, the 1-year-post-surgery mean TBS significantly improved by 12.12% ± 9.34, leading to a significant decrease in the T-gap to −0.296 ± 1.03. At two years post-surgery, we observed that the mean change in the BMD between 1 and 2 years post-surgery was minimal, with a mild decrease in the mean TBS and TBS-adjusted T-score and a mean T-gap of −0.44 ± 0.85, still much lower than the value at baseline. 

A graph of the individual T-gaps over time is presented in Figure 1. To validate the robustness of our results, we performed the same analysis including only the subset of participants with both BMD and TBS measurements in all three time points (N = 25), with similar results (data not shown).

To expose the potential causes for this dramatic T-gap, we performed correlation analyses with various pre-surgery anthropometric parameters. Overall body weight was negatively correlated with the T-gap (r = −0.33, *p* = 0.03), with only a trend for the correlation with BMI (r = −0.26, *p* = 0.09) and no correlation with body fat percentage (*p* = 0.42). The strongest correlation for the T-gap was that with waist circumference (r = −0.43 *p* = 0.004). The detailed correlation graphs with regression lines can be seen in Figure 2.

The closing T-gap from baseline to 1 and 2 years was thus explained by changes in weight circumference rather than changes in BMI. The changes in the mean WC and BMI at baseline and at 1 and 2 years after SG are shown in Table 5. 

Despite the lower T-gap at one year compared to that at baseline, the gap remained strongly correlated with WC at one year after SG (Figure 3). 

## 4. Discussion

We previously reported a decrease in total hip BMD but a stable LS BMD at 1 and 2 years after SG [12], compatible with the literature showing a consistent decrease in total hip BMD but not in LS BMD after bariatric surgery [7]. The impact of BS on bone microarchitecture has scarcely been evaluated, and few data have been published concerning the use of TBS in this context, despite it being a simple way to capture microarchitectural changes not seen in BMD.

In our present study, we analyzed TBS and TBS T-scores obtained from LS DXA at baseline and at 1 and 2 years after SG. The most striking result was the significantly lower TBS T-score relative to the BMD T-score pre-surgery, suggesting a partially degraded LS bone microarchitecture at baseline (mean TBS 1.25 ± 0.12). A finding that almost completely resolved with normal microarchitecture 1 year post-surgery (mean TBS 1.41 ± 0.08) was maintained at 2 years after SG (mean TBS 1.36 ± 0.10). In parallel, the LS BMD remained stable. It seems implausible for such an increase in the TBS to reflect true biological change; rather, we believe it stems from technical aspects of TBS measurement. The current guidelines recommend TBS measurement up to a BMI of 37 Kg/m^2^. At baseline, all our patients had a BMI close to or above 37 kg/m^2^, thus warranting caution in TBS interpretation. In fact, rather than BMI or total fat mass, waist circumference was more strongly negatively associated with the T-gap. This may suggest that the repartition of fat mass has an impact on TBS calculation, as increased waist circumference reflecting visceral fat mass in the pelvis may interfere with image acquisition due to increased tissue thickness, the latter impacting on TBS calculation. Obviously, a WC of 122 ± 10 cm (a mean WC at baseline, Table 1) warrants a cautious interpretation of the TBS, whereas 96 ± 11.9 cm is suitable. The threshold above which the TBS should not be interpreted would thus be in-between these two values. This concept is compatible with previous reports which showed that artificially increased tissue thickness in TBS phantoms is associated with a decreased measured TBS [16,17]. As a result, the TBS algorithm used currently takes into account tissue thickness via BMI adjustment and is thus strongly impacted by variations in weight [18], especially when using a Hologic manufacturer DXA machine [19]. An updated TBS algorithm, version 4.0, which directly accounts for tissue thickness, has overcome this limitation in several studies showing the negative correlation between TBS and BMI [20,21].

The limitations of our study include its inherent applicability to small cohorts, limiting the strength of the evidence and the potential for replicability. Moreover, we had no measure of visceral abdominal fat mass to prove the exact inverse correlation between waist circumference, visceral abdominal fat mass and TBS. However, this prospective report includes a typical population of patients subjected to bariatric surgery and, according to our knowledge, is the first to describe TBS changes after SG over two years of follow-up. 

## 5. Conclusions

We can learn from our study that at 1 and 2 years after SG, microarchitecture LS, as measured using the TBS, is normal in most patients and partially degraded in 12.5% and 25%, respectively. Thus far, only one published study has used the TBS in the context of bariatric surgery, specifically after RYGB, and described a stable normal TBS at one and three years after surgery [10]. The TBS should be interpreted cautiously at baseline before bariatric surgery, as well as in all patients with morbid obesity and a BMI above 37 kg/m^2^, but probably more so in those with elevated waist circumference, because tissue thickness impairs the validity of the TBS calculation. This issue may be better approached using a new TBS algorithm implementing the tissue thickness variable instead of BMI. Further studies evaluating the impact of bariatric surgery on hip microarchitecture are warranted.

## Figures and Tables

**Figure 1 nutrients-15-03481-f001:**
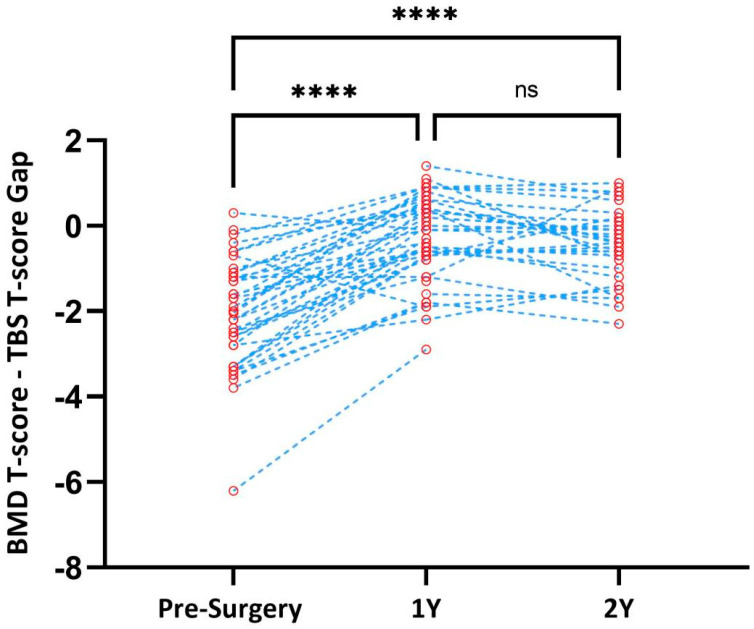
Differences between T-scores derived for lumbar spine bone mineral density and the trabecular bone score over time. *p*-values for mixed-effects analysis with Holm–Šídák’s multiple comparisons. BMD: bone mineral density; ns: non-significant; TBS: trabecular bone score. **** Statistically significant.

**Figure 2 nutrients-15-03481-f002:**
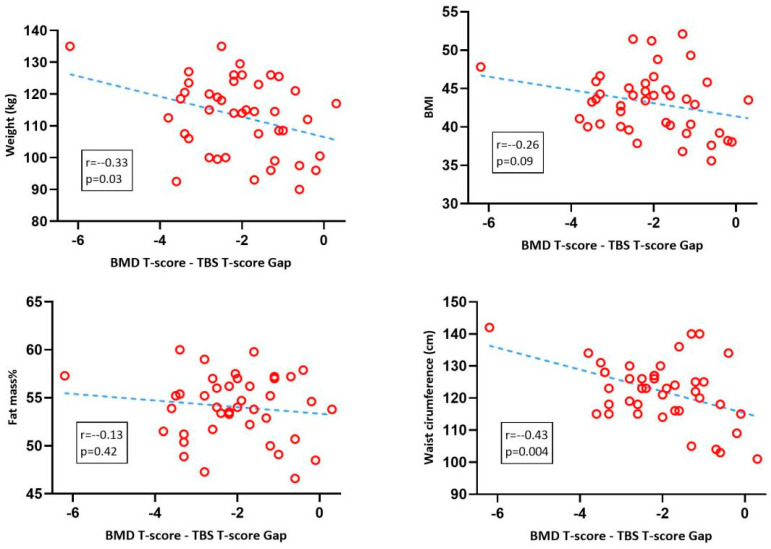
Pearson’s correlations. Blue dotted line—simple linear regression. BMD: bone mineral density; TBS: trabecular bone score.

**Figure 3 nutrients-15-03481-f003:**
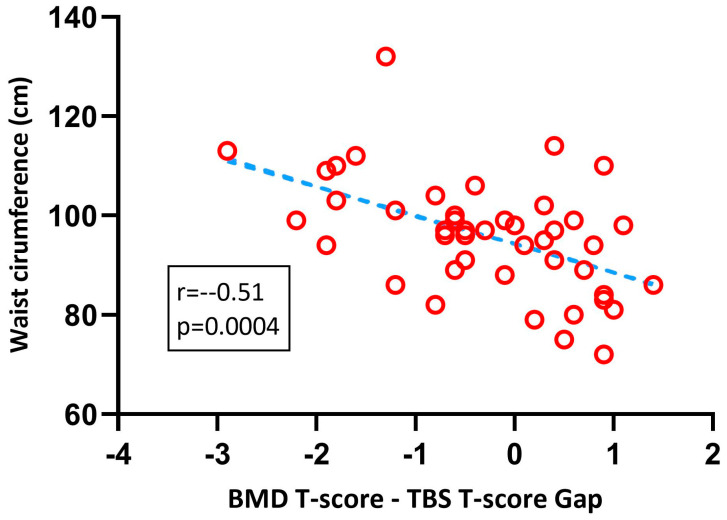
Pearson’s correlation between waist circumference and BMD–TBS T-score gap at 1 year post-surgery. BMD: bone mineral density; TBS: trabecular bone score.

**Table 1 nutrients-15-03481-t001:** Clinical characteristics of the included patients at baseline (n = 52 patients).

	Mean	SD
Age at surgery (years)	33.05	11.05
Height (cm)	161.4	4.86
Weight (kg)	112.80	12.11
BMI	43.27	4.06
Body fat mass%	54.54	3.33
Waist circumference (cm)	122.77	10.72

BMI: body mass index.

**Table 2 nutrients-15-03481-t002:** Classification of patients based on BMD WHO criteria and TBS.

	BMD T < −2.5	−2.5 < BMD T< −1.0	BMD T > −1.0	TBS < 1.2	1.2 < TBS < 1.3	TBS > 1.3
Baseline	1/52 (1.9%)	8/52 (15.4%)	43/52 (82.7%)	9/42 (21.5%)	17/42 (40.5%)	16/42 (38.0%)
1 year	2/52 (3.8%)	8/52 (15.4%)	42/52 (80.8%)	0/48 (0.0%)	6/48 (12.5%)	42/48 (87.5%)
2 years	1/31 (3.2%)	3/31 (9.7%)	27/31 (64.5%)	0/31 (0.0%)	8/31 (25.8%)	23/31 (74.2%)

BMD: bone mineral density; TBS: trabecular bone score.

**Table 3 nutrients-15-03481-t003:** Categories of patients according to BMI at baseline, 1 and 2 years after SG.

	Number of Patients with BMI > 37 kg/m^2^	Number of Patients with 35 < BMI < 37 kg/m^2^	Number of Patients with BMI < 35 kg/m^2^
Baseline	49/52 (94.2%)	3/52 (5.7%)	0/52 (0.0%)
1 year	6/52 (11.5%)	6/52 (11.5%)	40/52 (76.9%)
2 years	630 (20%)	3/30 (10%)	21/30 (70%)

BMI: body mass index.

**Table 4 nutrients-15-03481-t004:** Changes in BMD, BMD T-score, TBS, TBS- T-score, T-gap.

	LS BMD	LS T-Score	TBS	TBS- T-Score	T-Gap
Baseline	1.018 ± 0.09	−0.4 ± 0.93	1.25 ± 0.12 (N = 42)	−2.3 ± 1.33	−2.05 ± 1.26
1 year	1.01 ± 0.1 (vs. baseline: −0.93% ± 3.37, *p* = 0.06)	0.33 ± 1.00	1.41 ± 0.08 (vs. baseline: +12.12% ± 9.34, *p* < 1.5 × 10^−10^)	−0.62 ± 0.98	−0.296 ± 1.03
2 years	0.95 ± 0.12 (vs. 1-year: −0.31% ± 4.23)	−0.67 ± 1.00	1.36 ± 0.10 (vs. 1-year: −1.78% ± 5.16, *p* = 0.06)	−1.15 ± 1.12	−0.44 ± 0.85

BMD: bone mineral density; LS: lumbar spine; T-gap: BMD T-score–TBS T-score gap; TBS: trabecular bone score. Results presented as mean ± SD. *p*-Values for matched two-sided Student’s *t*-test.

**Table 5 nutrients-15-03481-t005:** Changes in BMI and WC at baseline and at 1 and 2 years after SG.

	BMI kg/m^2^ (Mean ± SD)	WC in cm (Mean ± SD)
Baseline	43.26 ± 4.1	122.7 ± 10
1 year	31.25 ± 4.8	95.9 ± 11.8
2 years	32.74 ± 4.7	96.7 ± 11.9

BMI: body mass index; SG: sleeve gastrectomy; WC: waist circumference.

## Data Availability

Not applicable.

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
