# Peer review of "Trabecular Bone Score Preceding and during a 2-Year Follow-Up after Sleeve Gastrectomy: Pitfalls and New Insights"

_nutrients, 2023, doi:10.3390/nu15153481_

Round 1
Reviewer 1 Report
The present study observed that the alterations of TBS in patients with SG. A few concerns about the submission.
1) The current title may not reflect the content. What is the meaning of baseline? Every experiment have controls and that should not be mentioned in the title.
2) The conclusion in the abstract needs to be clearly presented.
3) Please pay attention to the rules of the abbreviations.
4) In line 104, the word "changs' should be 'changes'.
5) The results in the table 3 should be briefly explained.
6) what is the main goal of the investigation?
Author Response
Dear Reviewer, thank you for your review of our manuscript.
We answered point-by-point to your comments:
1) The title was slightly modified, and we erased the term "Baseline".
It is now: Trabecular Bone Score Preceding and During 2-year Follow-up after Sleeve Gastrectomy: Pitfalls and New Insights
2) We structured and labelled the whole abstract including a background-context and the hypothesis of the study, then the measures we used, the results, the interpretation and the conclusions. The latter starts now with: In conclusion....
3) We checked all the abbreviations in the text, figures and tables.
So in Table 1 and 5 we made appropriate changes, and we added abbreviations for BMI and TBS in Figures 1,2 and 3, and made some modifications in Table 5.
All over the text, we made also requested changes concerning the abbreviations.
4) The typo was corrected.
5) We added an explanative sentence in the corresponding paragraph preceding Table 3:
Obviously, before SG, most patients had BMI > 37 kg/m2, which decreased significantly at one and 2 years after SG; they represented only 20 % of patients at the end of the follow-up. Between the first and second year, however, this proportion mildly increased.
6) The main goal of the investigation was added both in the abstract (To test the hypothesis that TBS is clinically useful for this population) and at the end of the introduction (The main goal of our current investigation was to evaluate the hypothesis that TBS has valuable clinical relevance in patients undergoing SG despite their elevated BMI before surgery, and that TBS may be impacted by SG during follow-up ).
Reviewer 2 Report
Stokar et al. analyzed bone parameters from 52 (full data from 31) patients before, one and two years after sleeve gastrectomy surgery (SG). The main findings are that one year post-surgery the bone parameters improved, and remained stable for 24 months. A second finding is that high abdominal fat in patients with high BMI may lead to lower trabecular bone score due to high abdominal fat that may interfere with image acquisition.
Both of these findings are clinically relevant and are of interest to the clinicians. Additional strengths of the manuscript are the rigor of the statistical, and correlation with potential confounding variables.
There are a few key weaknesses that, if addressed, would improve the manuscript.
1) The terminology used is confusing and the acronyms are not clearly defined to ensure that they are used consistently throughout the manuscript.
2) The study, as presented is, and should be clearly labeled as a longitudinal cohort design.
3) The abstract is difficult to read and can be organized better and labeled as to what was measured in the cohort, the findings and the interpretation.
4) The introduction does not clearly state a hypothesis, or the central question addressed. While the introduction is adequate, it can be improved by editing.
5) [Minor point] The Holm-Sidak correction is unusual for multiple comparison corrections. Can the authors justify the use of this correction choice?
The quality of English language is adequate (with respect to grammar), the manuscript is a bit disorganized and can be improved by editing.
Author Response
Dear Reviewer, thank you for your review of our manuscript.
We answered point-by-point to your comments:
1) The main issue concerning the terminology in the manuscript was in our view the BMD-T Score- TBS-T score gap. We renamed it: T-gap. This terminology is more friendly-reader, all over the manuscript and in the abstract and we defined it in the Methods section. We only remained it as is in the legends of figures.
TBS results were associated with a TBS-derived T – score allowing us to compare the latter with the DXA-related LS T-score and to calculate a BMD T-Score -TBS T-score gap ( T-gap).
2) we corrected this appropriately, and it now appears both in the abstract and the introduction that it is a Longitudinal Cohort Study.
3)We labelled more clearly the abstract with a background sentence, a hypothesis to test, the measures over the study, and the results at baseline, one and 2 years. Finally, interpretation and conclusions.
4) we added a clear statement about the hypothesis and main goal of the study at the end of the introduction:
The main goal of our investigation was to evaluate the hypothesis that TBS has valuable clinical relevance in patients undergoing SG despite their elevated BMI before surgery, and that TBS may be impacted by SG during follow-up.
5) As we compared selected pairs of means valued we had the option of using the Tuckey test or the Holm-Sidack test... Both were available with our statistic program, but as with did not want to compute confidence intervals, our program automatically used the Holm Sidack test. We obtained similar results with a Tuckey test...
6) We made all over the manuscript some english-language editions.